# Echocardiographic Parameters of *Callithrix* spp. Under Human Care

**DOI:** 10.3390/ani15131875

**Published:** 2025-06-25

**Authors:** Melina Castilho de Souza Balbueno, Jessica Amancio Martins, Soraya Kezam Malaga, Ralph Eric Thijl Vanstreels, Cideli de Paula Coelho

**Affiliations:** 1Department of One Health Master’s Degree Program, University of Santo Amaro, Rua Professor Enéas de Siqueira Neto 340, Jardim das Imbuias, São Paulo 04829-300, SP, Brazil; mecastilho3@yahoo.com.br (M.C.d.S.B.); jehamanciovet@gmail.com (J.A.M.); smalaga@uol.com.br (S.K.M.); 2HD Science School, Rua São Paulo, 1096—Santa Paula, São Caetano do Sul 09541-100, SP, Brazil; 3Laboratory of Comparative Wildlife Pathology, Department of Pathology, School of Veterinary Medicine and Animal Science, University of São Paulo, Av. Prof. Dr. Orlando Marques de Paiva 87, Cidade Universitária Armando Salles de Oliveira, São Paulo 05508-270, SP, Brazil; ralph_vanstreels@yahoo.com.br

**Keywords:** atrial diameter, echocardiogram, heart rate, marmoset, non-human primates, shortening fraction, ventricular diameter

## Abstract

This study provides echocardiographic parameters for marmosets of the genus *Callithrix* under human care, including species such as *Callithrix aurita*, *C. jacchus*, *C. penicillate*, and hybrids. The aim was to establish reference values by using Doppler ultrasound, through assessing parameters like ventricular and atrial diameters, shortening fraction, and ejection fraction. In some animals, mild valvular insufficiencies were observed. These data serve as an initial refence for monitoring cardiac health in marmosets, thus aiding in both the conservation and management of captive primates.

## 1. Introduction

Non-human primates (NHPs) are frequently used in anatomical and physiological research due to their similarity to humans [1], and as experimental models for diseases [2]. However, the lack of studies on several species of marmosets contributes to difficulties regarding their conservation, both under human care and in the wild.

Callitrichids can present arrhythmias and murmurs, which can be audible during physical examination, as well as myocardial fibrosis and arteriosclerotic lesions [3]. Furthermore, heart disease is a leading cause of death in many primates living under human care [4].

Doppler echocardiography is an important diagnostic tool for heart disease, and parameters for some species of animals, including dogs, cats, and horses, among others, are already well established. With technological advancements, the possibility of having resources of greater precision for improving diagnoses, such as tissue Doppler, strain, and three-dimensional imaging, has emerged [5]. However, for some animal species, the normal reference values have not yet been well defined.

This study was the first to evaluate echocardiographic parameters in *Callithrix* spp., kept under human care in captivity in a non-governmental organization (NGO) and a zoo. Its aim was to establish echocardiographic parameters across different species and age groups, which ranged from 0.4 to 15 years.

## 2. Materials and Methods

A total of 168 marmosets were included in the study, comprising 30 *C. penicillata*, 20 *C. aurita*, 67 *C. jacchus*, and 51 hybrids, of both sexes, ideal body condition, with an average weight of 328 ± 71 g and range 0.4 to 15 years (Table 1). These animals were kept under human care at the Mucky Project in Itu and in the São Bernardo do Campo Zoo, both located in the state of São Paulo, Brazil.

Samples were collected between 2 November 2021 and 16 May 2022. The procedures were authorized through UNISA’s CEUA no. 57/2021 and SISBIO no. 78874-1.

The sampling of both species was carried out by convenience. The inclusion criteria for both genders were animals belonging to the selected species and that accepted handling and contact by the keepers. Primates that were not fasted prior to sedation or that were not clinically healthy were excluded.

The animals were caught by the keepers using manual restraint. Subsequently, the animals were sedated with isoflurane via mask induction and this was maintained at a rate of 1 to 3% with 100% oxygen for the duration of the examination. Doppler echocardiographic examinations were performed using a MyLab Gamma veterinary ultrasound machine (Esaote, Milan, Italy), with the animal in dorsal recumbency, using a foam pad for positioning support. The average time was 15 min for each animal, with a maximum duration of 20 min.

The sectorial transducer used was a P2 (pediatric) model with a frequency range of 3–11 MHz. Conductive gel was applied to ensure contact between the transducer and the patient’s skin, without prior trichotomy to obtain the images. Settings for gain and depth were adjusted individually for each animal to achieve the optimal image quality.

The transducer was positioned between the 2nd and 5th intercostal spaces on both the right and left hemithorax, in the parasternal window, apical or subcostal thoracic regions (Figure 1A,B).

For the right-side view, the assessment was started in the parasternal long-axis window. The transducer was then rotated to obtain a short-axis image. Measurements were taken of the left ventricular free wall and interventricular septal thickness in diastole and systole (LVPWd, LVPWs, IVSd, and IVSs), end-diastolic and end-systolic left ventricular diameter (LVIDd and LVIDs) in M-mode (Figure 2), aortic root diameter (Ao) and left atrium (LA), in two-dimensional mode, and the end-diastolic ratio between the aorta and left atrium (LA/Ao) were derived. The shortening fraction (FS) and ejection fraction (EF; Teicholz method) were estimated. Additional measurements included the septal separation of the mitral E-point (EPSS) and pulmonary artery flow velocity (P Vmax).

The end of the systole and diastole were related to the cardiac cycle using a coupled electrocardiogram.

The echocardiographic examination and standard measurements were performed according to previously established protocols for non-human primates [6,7] and according to the Guidelines for performing transthoracic echocardiographic examination for humans, American Society of Echocardiography [8].

The parameters were assessed in B-mode and M-mode, with color and spectral Doppler.

In the left parasternal and subcostal windows, the apical 4-chamber section was imaged, in B-mode. The 4-chamber images were assessed both longitudinally and apically for subjective analysis (Figure 3).

This view also allowed assessment of mitral early diastolic flow (E wave), mitral late diastolic flow (A wave), apical 5-chamber flow, peak aortic velocity (Ao Vmax), isovolumetric relaxation time (IVRT), and the E:IVRT ratio.

The flow of the atrioventricular, mitral, tricuspid, semilunar, aortic, and pulmonary valves, as well as those in the great vessels, were measured using color and spectral Doppler for each valve. Higher pulse repetition frequency (PRF) settings were used to avoid aliasing in normal flow.

Heart rate (HR) was obtained using the transverse image of the left ventricle (short axis) in M-mode from the right parasternal window, and 3 systolic cycles were evaluated for each patient.

The mean, standard deviation (SD), first quartile, median, third quartile, minimum and maximum values were calculated to describe the distribution of the results. Shapiro−Wilk tests were used to determine if data were normally distributed. Breusch−Pagan tests were used to verify data homoscedasticity. Two-way analysis of variance (ANOVA), followed by Tukey’s post hoc test, was used to determine whether the age, body mass and echocardiographic parameters differed according to species and sex. The significance level was set at 0.05 for all tests.

## 3. Results

The echocardiographic parameters (mean ± SD) for *Callithrix aurita*, *C. jacchus*, *C. penicillate*, and hybrids (referred to as *Callithrix* sp.) were evaluated and are presented in Table 2.

Figure 4 illustrates the distribution of the echocardiographic parameters where differences were observed both between species and between the sexes; the heart rate, aortic root diameter, left ventricular diameter in systole and in diastole, left ventricular wall thickness in systole, and isovolumetric relaxation time were different.

Among the echocardiographic alterations diagnosed, mild mitral valve regurgitation accounted for 14 of cases, and 7 animals with mild tricuspid valve regurgitation were observed.

## 4. Discussion

This study was the first to assess echocardiographic parameters in *Callithrix* spp. Differences were observed in several measurements, including in the aortic root, diameter, and left ventricular diameter in the systole and diastole, particularly in males, between *C. aurita* and *C. jacchus*. Although the average weight of the animals is similar, the stature of *C. aurita* differs from that of *C. jacchus*, which is smaller in stature, and this may influence the parameters [9].

Doppler echocardiographic measurements have previously been taken on 9 specimens of spider monkeys (*Ateles* sp.) [10], 16 capuchin monkeys (*Cebus apella*, Linnaeus, 1758) [11], 247 specimens of cynomolgus monkeys (*Macaca fascicularis*) [12], and 823 Rhesus monkeys (*Macaca mulatta*) [13].

In Rhesus monkeys, the average value of the left ventricle diameter in the end-diastole was 2.18 cm ± 0.35 and in the end-systole it was 1.33 cm ± 0.29 [13]; in capuchin monkeys was 1.37 cm ± 0.33 and 0.99 cm ± 0.28 in the diastole and systole [11], respectively; in spider monkeys, the normal mean was 2.24 cm ± 0.18 in diastolic phase and 1.41 cm ± 0.25 in the systole [10]; in cynomolgus monkeys, the mean was 1.63 cm ± 0.34 and 1.03 cm ± 0.26 in the diastole and systole in M-mode [12]; while in this study, in marmosets, the average values were 0.68 cm ± 0.08 and 0.39 cm ± 0.06 in the diastole and systole, respectively.

In capuchins, the normal mean of the left atrium diameter was 0.74 cm ± 0.15 and the aortic root diameter was 0.62 cm ± 0.12 [11]; in cynomolgus monkeys, the left atrium was 1.33 cm ± 0.26 and 0.87 cm ± 0.19 [12]; while in *Callithrix* spp., the overall mean observed was 0.36 cm ± 0.04 and the left atrium was 0.47 cm ± 0.05. These parameters are related to the weight of the animal. The weight of Rhesus was 8.49 kg ± 3.52 [13]; in spiders it was 8.15 kg ± 0.14 [12]; in capuchins it was 1.95 kg ± 0.40 [11]; in cynomolgus monkeys was 4.1 kg ± 1.31 [12]; while marmosets weighted 328 g ± 7.

The velocity of the E wave was 0.72 m/s ± 0.20 and the A wave was 0.54 m/s ± 0.17 in cynomolgus monkeys [12]. In capuchins, the mean E wave was 0.76 m/s ± 0.15 and the A wave was 0.44 m/s ± 0.10 [11]; in the present study, the E wave was 0.57 m/s ± 0.17 and the A wave was 0.41 m/s ± 0.23.

A previous electrocardiogram study evaluated 19 healthy adult *C. penicillata* and established parameters among individuals anesthetized with tiletamine and zolazepam, where the average HR was 264 ± 74 bpm [14]. In the present study, only isoflurane was used for anesthesia and echocardiography was performed in these animals, and the HR was similar to that observed in the earlier study, i.e., 276.86 ± 45.89 bpm.

In a study using magnetic resonance imaging that assessed ventricular function in marmosets, it was found that heart weight and heart rate tended to increase with age, while body weight appeared to decrease slightly. The mean ejection fraction in that study was 56 ± 7% (ranging between 41.5 and 69.0%) [15]. In the present study, the ejection fraction measured through echocardiography using the Teicholz method was 78.61 ± 6.48% on average: 78.1 ± 6.5% in hybrids, 78.4 ± 6.9% in *C. penicillata*, 79.7 ± 6.4% in *C. jacchus* and 76.3 ± 5.6% in *C. aurita*.

In the present study, no relationship was observed in the diagnosis of alterations in cardiological exams with advancing age, which ranged from 0.4 to 15 years, with an average of 5.3 ± 3.2 years. In another study, cardiovascular findings at necropsy accounted for 25% of *C. jacchus* [16]. Although the current study did not aim to correlate post mortem findings with echocardiographic alterations, 80% of *Callithrix* spp. did not present any alterations in the echocardiogram. The main alterations observed were mild mitral valve insufficiency and mild tricuspid valve insufficiency, with no evidence of hemodynamic repercussions in these animals. Additionally, the animals in the study consumed a balanced diet and the animals received veterinary care when necessary. However, those that were not clinically healthy were excluded from this study. None of the animals were treated using heart medication.

Some anesthetics can depress the myocardium and alter echocardiogram parameters. One study observed hypoxemia under two protocols in common marmosets, one group being the administration of ketamine, xylazine, and atropine-associated and the other with alfaxalone [17]. The use of Telazole in 10 *C. jacchus* caused a reduction in heart rate over time, and reduced the respiratory rate in both protocols [18]. Thus, the present study opted for the use of isoflurane for the sedation of the animals. Studying isoflurane revealed the occurrence of a good anesthetic plan during the procedure, in addition to low complication rates throughout the surgical period [19].

After completing the study, a 15-year-old female *C. penicillata* presented to the veterinarian abdominal distension and anorexia. During the subsequent physical examination, a murmur, gallop rhythm, and ascites were observed. An echocardiogram was performed and revealed systolic dysfunction with enlarged left chambers, a shortening fraction of 21% (normal mean: 42 ± 6%), and an ejection fraction of 47% (normal mean: 78 ± 6%), leading to a diagnosis of dilated cardiomyopathy phenotype based on the echocardiogram findings. This was the first reported case in this species, and it was confirmed post-mortem through necropsy [20]. However, this animal was not part of the sample for this study—it was assessed and reported after the end of data collection.

## 5. Conclusions

This study established echocardiographic parameters in *Callithrix* spp. (*C. aurita*, *C. jacchus*, *C. penicillata* and hybrids) that can serve as a reference for diagnostic purposes and future research.

## Figures and Tables

**Figure 1 animals-15-01875-f001:**
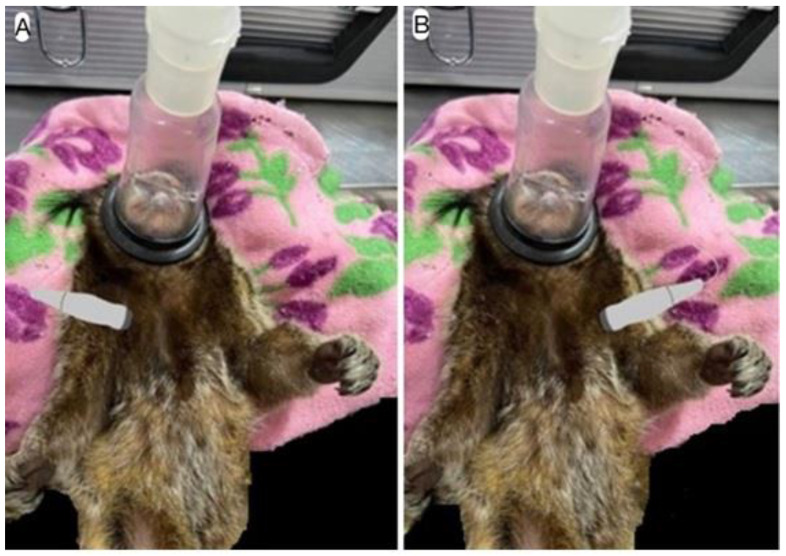
Schematic image of the access window for Doppler echocardiography images: (**A**) right hemithorax window and (**B**) left hemithorax window.

**Figure 2 animals-15-01875-f002:**
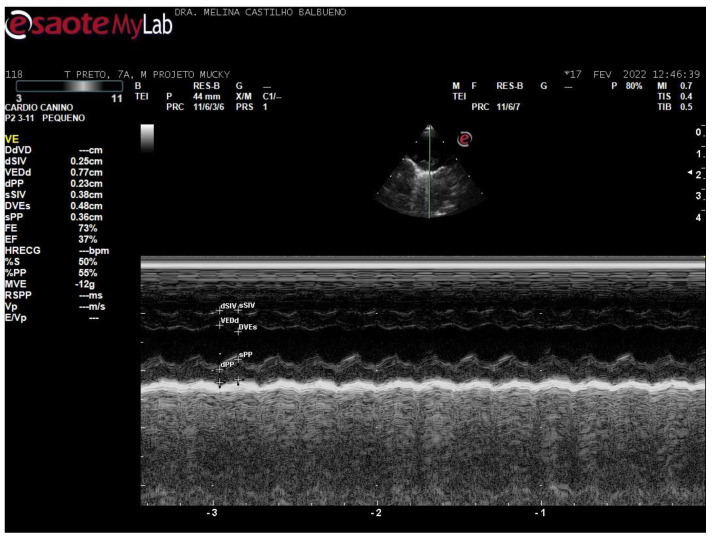
Cross-sectional Doppler echocardiographic image in M mode, with the following measurements: interventricular septum, ventricular diameter, ventricular wall in systole and diastole, shortening fraction and ejection fraction, from a 7-year-old male *Callithrix penicillata*.

**Figure 3 animals-15-01875-f003:**
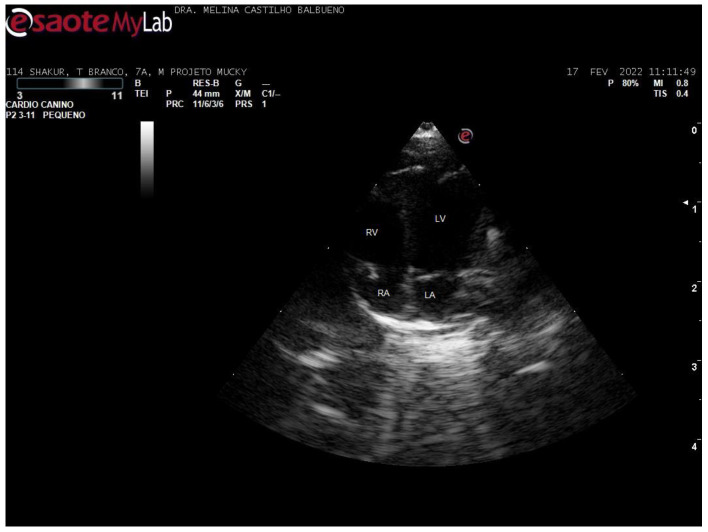
Apical 4-chamber echocardiographic image of a 7-year-old male *Callithrix jacchus*.

**Figure 4 animals-15-01875-f004:**
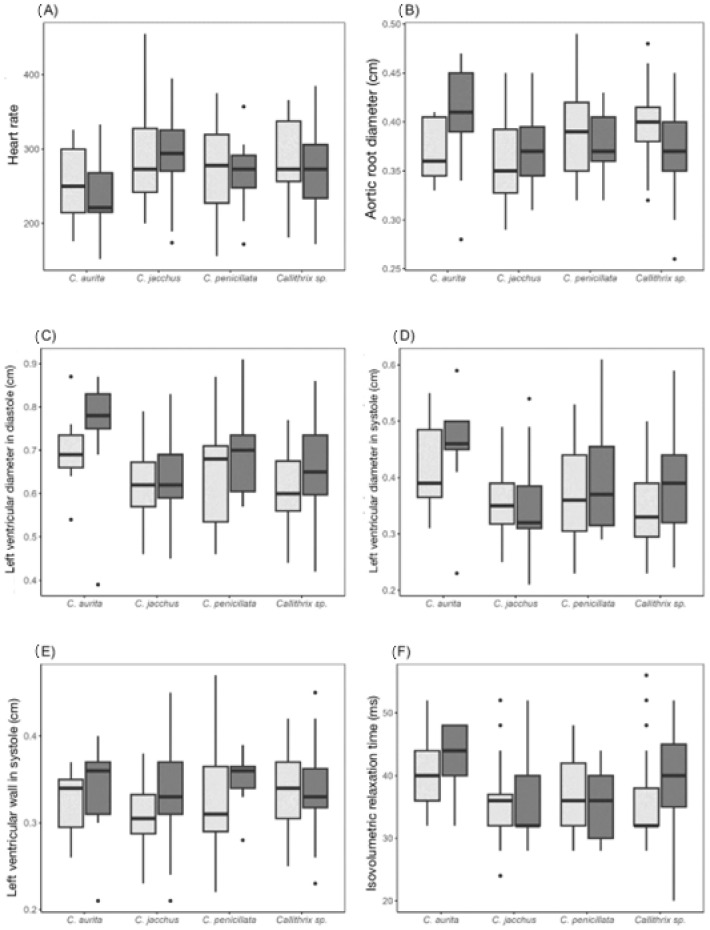
Diagrams of heart rate and echocardiographic parameters of marmosets (*Callithrix* sp.) in which there are a statistical difference between species and/or the sexes. Females are represented in light grey and males in dark grey.

**Table 1 animals-15-01875-t001:** Descriptive number of animals according to sex and age.

Animals						
Sex		Age Groups				
F	M	0.4–2 y	3–5 y	6–8 y	9–11 y	12/15 y
77	91	57	26	63	15	7

**Table 2 animals-15-01875-t002:** Descriptive statistics of the heart rate and echocardiographic parameters of marmosets (*Callithrix* spp.) kept on isoflurane during the examination. The results are presented as mean ± standard deviation {first quartile, median, third quartile} [minimum–maximum]. The normal distribution refers to the Shapiro−Wilk test (significance level = 0.05). The *p* values for the analysis of variance (ANOVA) in relation to species (“*P_species_*”) and sex (“*P_sex_*”) are included, and asterisks are used to highlight cases where there was significance. Different superscript letters indicate the presence of a significant difference (Tukey’s test) between the species.

Parameter	Normal Distribution	*C. aurita*	*C. jacchus*	*C. penicillata*	*Callithrix* sp.	*P_species_*	*P_sex_*
Heart rate (bpm)	Yes	242.2 ± 50.3 {212, 232.5, 283.3} [152–333] ^a^	289.7 ± 55.5 {248, 288, 326} [174–455] ^b^	270.1 ± 59.6 {233.8, 278, 313} [156–375] ^ab^	278.4 ± 54.8 {246.5, 273, 316} [172–385] ^ab^	0.009 *	0.504
Aortic root diameter (cm)	No	0.39 ± 0.05 {0.35, 0.4, 0.43} [0.28–0.47] ^a^	0.36 ± 0.04 {0.33, 0.36, 0.4} [0.29–0.45] ^b^	0.39 ± 0.04 {0.35, 0.39, 0.42} [0.32–0.49] ^ab^	0.38 ± 0.04 {0.36, 0.39, 0.4} [0.26–0.48] ^a^	0.007 *	0.989
Left atrium diameter (cm)	No	0.48 ± 0.06 {0.44, 0.49, 0.54} [0.37–0.59] ^a^	0.47 ± 0.06 {0.42, 0.48, 0.51} [0.33–0.61] ^a^	0.5 ± 0.05 {0.47, 0.49, 0.53} [0.41–0.61] ^a^	0.48 ± 0.06 {0.45, 0.48, 0.52} [0.35–0.62] ^a^	0.216	0.482
Ratio of the left atrial dimension to the aortic annulus dimension	Yes	1.24 ± 0.12 {1.15, 1.21, 1.32} [1.06–1.44] ^a^	1.3 ± 0.13 {1.22, 1.31, 1.37} [1.02–1.6] ^a^	1.3 ± 0.13 {1.21, 1.3, 1.37} [1.04–1.61] ^a^	1.26 ± 0.14 {1.15, 1.27, 1.38} [0.98–1.5] ^a^	0.154	0.472
Interventricular septum thickness at end-diastole (cm)	No	0.22 ± 0.02 {0.2, 0.22, 0.23} [0.16–0.26] ^a^	0.21 ± 0.03 {0.19, 0.21, 0.23} [0.14–0.29] ^a^	0.21 ± 0.03 {0.19, 0.21, 0.23} [0.16–0.28] ^a^	0.2 ± 0.03 {0.17, 0.2, 0.22} [0.15–0.28] ^a^	0.129	0.057
Interventricular septum thickness at end-systole (cm)	No	0.33 ± 0.05 {0.3, 0.32, 0.37} [0.23–0.44] ^a^	0.33 ± 0.04 {0.3, 0.32, 0.36} [0.24–0.45] ^a^	0.34 ± 0.05 {0.3, 0.34, 0.38} [0.26–0.46] ^a^	0.33 ± 0.05 {0.3, 0.33, 0.36} [0.22–0.47] ^a^	0.754	0.075
Left ventricular internal dimension at end-diastole (cm)	No	0.74 ± 0.12 {0.69, 0.76, 0.81} [0.39–0.87] ^a^	0.63 ± 0.09 {0.58, 0.62, 0.69} [0.45–0.83] ^b^	0.66 ± 0.11 {0.59, 0.68, 0.72} [0.46–0.91] ^ab^	0.64 ± 0.1 {0.57, 0.63, 0.72} [0.42–0.86] ^b^	<0.001 *	0.024 *
Left ventricular internal dimension at end-systole (cm)	No	0.44 ± 0.08 {0.41, 0.46, 0.5} [0.23–0.59] ^a^	0.35 ± 0.06 {0.31, 0.33, 0.39} [0.21–0.54] ^b^	0.38 ± 0.09 {0.3, 0.37, 0.45} [0.23–0.61] ^b^	0.37 ± 0.08 {0.32, 0.36, 0.41} [0.23–0.59] ^b^	<0.001 *	0.077
Left ventricular posterior wall thickness at end-diastole (cm)	No	0.21 ± 0.02 {0.2, 0.21, 0.22} [0.15–0.24] ^a^	0.2 ± 0.02 {0.18, 0.2, 0.21} [0.15–0.25] ^a^	0.21 ± 0.04 {0.18, 0.2, 0.22} [0.15–0.33] ^a^	0.2 ± 0.04 {0.17, 0.21, 0.22} [0.11–0.3] ^a^	0.539	0.440
Left ventricular posterior wall thickness at end-systole (cm)	No	0.33 ± 0.05 {0.31, 0.34, 0.37} [0.21–0.4] ^a^	0.32 ± 0.05 {0.3, 0.32, 0.36} [0.21–0.45] ^a^	0.33 ± 0.05 {0.3, 0.34, 0.37} [0.22–0.47] ^a^	0.34 ± 0.04 {0.31, 0.33, 0.37} [0.23–0.45] ^a^	0.149	0.048 *
E-point-to-septal separation (cm)	No	0.08 ± 0.02 {0.07, 0.09, 0.09} [0.05–0.11] ^a^	0.09 ± 0.02 {0.07, 0.08, 0.1} [0.06–0.16] ^a^	0.09 ± 0.01 {0.08, 0.09, 0.09} [0.07–0.11] ^a^	0.09 ± 0.02 {0.08, 0.08, 0.09} [0.06–0.15] ^a^	0.872	0.334
Aortic flow (m/s)	No	0.66 ± 0.19 {0.57, 0.69, 0.75} [0.33–1.02] ^a^	0.77 ± 0.23 {0.65, 0.74, 0.9} [0.33–1.85] ^a^	0.73 ± 0.19 {0.62, 0.72, 0.84} [0.4–1.26] ^a^	0.69 ± 0.2 {0.52, 0.7, 0.8} [0.36–1.13] ^a^	0.069	0.295
Pulmonary flow (m/s)	No	0.66 ± 0.19 {0.52, 0.6, 0.81} [0.41–1.06] ^a^	0.65 ± 0.2 {0.5, 0.61, 0.78} [0.29–1.19] ^a^	0.67 ± 0.23 {0.47, 0.59, 0.86} [0.37–1.17] ^a^	0.63 ± 0.23 {0.46, 0.63, 0.75} [0.29–1.25] ^a^	0.877	0.158
Aortic flow gradient (mmHg)	No	1.85 ± 1 {1.33, 1.9, 2.23} [0.4–4.1] ^a^	2.41 ± 1.14 {1.7, 2.2, 3.2} [0.4–5.3] ^a^	2.26 ± 1.21 {1.5, 1.9, 2.8} [0.7–6.4] ^a^	2.05 ± 1.11 {1.1, 1.9, 2.55} [0.5–5.1] ^a^	0.165	0.133
Pulmonary flow gradient (mmHg)	No	1.9 ± 1.11 {1.08, 1.4, 2.65} [0.7–4.5] ^a^	1.82 ± 1.11 {1, 1.5, 2.45} [0.3–5.7] ^a^	2 ± 1.38 {0.9, 1.35, 2.98} [0.5–5.5] ^a^	1.79 ± 1.29 {0.9, 1.6, 2.25} [0.3–6.2] ^a^	0.885	0.157
Shortening fraction (%)	Yes	39.7 ± 4.2 {37, 40.5, 43} [33–47] ^a^	43.6 ± 5.9 {40, 44, 48} [30–58] ^a^	42.6 ± 6.8 {36.3, 42.5, 48} [33–57] ^a^	42.4 ± 6.2 {38, 42, 47} [28–54] ^a^	0.086	0.812
Ejection fraction (%)	No	76.3 ± 5.6 {72.8, 76.5, 80} [67–88] ^a^	79.7 ± 6.4 {76.5, 81, 84.5} [62–92] ^a^	78.4 ± 6.9 {72, 79.5, 84} [67–91] ^a^	78.1 ± 6.5 {74, 78, 84} [60–89] ^a^	0.176	0.835
E wave—Initial mitral diastolic flow (m/s)	Yes	0.49 ± 0.21 {0.4, 0.52, 0.66} [0.05–0.8] ^a^	0.57 ± 0.15 {0.47, 0.6, 0.67} [0.3–0.88] ^a^	0.59 ± 0.17 {0.48, 0.54, 0.68} [0.34–0.99] ^a^	0.59 ± 0.18 {0.46, 0.56, 0.67} [0.3–1.24] ^a^	0.145	0.704
A wave—Mitral late diastolic flow (m/s)	No	0.42 ± 0.16 {0.32, 0.39, 0.49} [0.14–0.7] ^a^	0.41 ± 0.23 {0.28, 0.43, 0.54} [0–0.97] ^a^	0.47 ± 0.22 {0.32, 0.47, 0.58} [0–1.08] ^a^	0.39 ± 0.27 {0.23, 0.36, 0.6} [0–1.01] ^a^	0.535	0.604
Ratio of E to A	No	1.28 ± 0.66 {0.83, 1.22, 1.66} [0.14–2.85] ^a^	1.3 ± 0.43 {1.07, 1.24, 1.58} [0.56–2.67] ^a^	1.32 ± 0.41 {0.94, 1.28, 1.66} [0.49–2.09] ^a^	1.3 ± 0.49 {0.96, 1.23, 1.54} [0.63–2.84] ^a^	0.995	0.161
Isovolumetric relaxation time (ms)	No	41.8 ± 5.7 {36, 44, 45} [32–52] ^a^	35.3 ± 6.5 {32, 36, 40} [24–52] ^b^	36 ± 5.9 {32, 36, 40} [28–48] ^b^	38.3 ± 8.1 {32, 36, 44} [20–56] ^ab^	0.003 *	0.424

## Data Availability

The data presented in this study are available on request from the corresponding author in the form of PhD research data.

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
