# Peer review of "Echocardiographic Parameters of *Callithrix* spp. Under Human Care"

_animals, 2025, doi:10.3390/ani15131875_

Round 1
Reviewer 1 Report
Comments and Suggestions for Authors
Dear,
Thank you for your submission ‘echocardiographic parameters of Callithrix spp under human care’. The obtained data are very valuable as reference values. However, some fine tuning of this manuscript is needed.
I have several comments:
Line 15: cardiac standards = ‘reference values’ or ‘reference ranges for’….. sounds better to me
Line 17: ‘Parameters considered normal for the species were observed, with only mild valvular insufficiencies in some cases’. Could be ‘Cardiac reference values were assessed. In some animals mild valvular insufficiencies were observed.
Abstract: normally an abstract contains background, study design, results and conclusions. Currently, to much useless information is provided like “The procedures were authorized through UNISA's CEUA no. 28 57/2021 and SISBIO no. 78874-1.”. Please reorganize the whole abstract. Add that you used clinical healthy animals. Write that you assessed cardiac reference values…… Currently no results are included. As line 31/32 is method and then discussion.
Keywords: Currently you have 5 from which 3 refer to marmoset/primate while you are allowed to use 10, so please add more like ventricular diameter, heart rate, isoflurane, aortic flow….. maybe delete captive primates
Introduction:
- The flow is missing: now is goes from model, lack of knowledge, Callitrichidae in brazil, heart disease, echo, goal. Maybe: Callitrichidae in brazil but also in biomedical research, healthy animals and reference values are needed……please adapt. And in line 63 you write that you will check age groups. In the abstract I miss that inclusion of age and sex as parameter.
Materials and methods:
Line 66: callithric penicillate can be C. Penicillate as in line 47 you already mentioned this species.
Line 66: clinical healthy animals? Sex? Weight? Body condition score so skinny or fat? Also describe the ages and age groups as in line 63 you wrote ‘age groups’ and in line 114 body mass.
Line 67: ‘..of both sexes and various age groups’. Maybe provide a table in which you show age groups, sex and bodymass with the number of animals in that group.
The order could be changed so: animals, how they were catched, sedated and then describe the position and ultrasound.
Some details are missing like:
- Maybe add that the animals were fasted the night before
- Maybe add in short how the animals were catched. Or were they trained to come and to be grabbed? Just to let the reader imagine if the animals were stressed or relaxed during the scan.
Figure 1: not sharp, no additional value to me. Can be deleted.
Line 127/128: The parameters were assessed in B-mode and M-mode (Fig. 3), with color and spec-127 tral Doppler. I think this should be moved to the material and method section as in that section the word B-mode is missing.
Line 133-137: ‘Abbreviations: EF: Ejection fraction; FS: Fractional shortening; IVSd: Interventricular 133 septum thickness at end-diastole; IVSs: Interventricular septum thickness at end-systole; 134 LVIDd: Left ventricular internal dimension at end-diastole; LVIDs: Left ventricular inter-135 nal dimension at end-systole; LVPWd: Left ventricular posterior wall thickness at end-136 diastole; LVPWs: Left ventricular posterior wall thickness at end-systole.’ I think this can be deleted as this is all mentioned in line 87-95, right?
Figure 4 is really hard to read, on paper and on a screen. Please adapt. And I assume that line 146-147 should be included in the figure legend.
Line 149: 9 should be nine
Line 154: 'found' should be more scientific so 'observed'
Line 159 C. penicillate and the word marmoset can be deleted.
Line 161. The , should be a . and start the new sentence with There
Line 173: C. jacchus
Line 179: C. penicillate
I miss in the discussion the influence of isoflurane compared to other anesthesics or unsedated. Please add how people should interpretate this data when they will use for example alfaxalone or do unsedated scans.
I miss also the discussion about the observed differences between age groups and sexes.
Reviewer 2 Report
Comments and Suggestions for Authors
Overall, this is a well written paper describing normal echocardiography parameters for multiple species of Callithrix, a common laboratory animal species. This information is relevant to establish normal baseline data. Common, well described, techniques were used to determine the parameters. Each section of the article is well written and easy to read.
This reviewer would like to make three recommendations. The first is to please include the actual age range of participants within the study. The second is associated with the charts/graphs and statistical analysis. It's clear that there is a statistical significance between species but it's not clear if we are comparing all species or if each species is being compared to C. aurita. Recommend lines on the chart to indicate where the significance lies between the two species. Third recommendation is with the last paragraph of the discussion. It is currently not clear if this animal was a participant in the study that later developed clinical signs or if this was completely unrelated event used to demonstrate the difference between normal and abnormal.
Reviewer 3 Report
Comments and Suggestions for Authors
A review of a paper "Echocardiographic parameters of Callithrix spp under human care".
Authors present a very interesting study of echocardiographic parameters in marmosets. The study was conducted on significant number of animals what is its high advantage. I think that the paper will serve to broaden scientific knowledge. Nonetheless, in my opinion, some corrections are required prior to acceptance of the paper.
Below, I present my concerns and comments:
- line 32, 149, 151: the term "specimen" is in my opinion not the best in that context; consider replacing the term, for example with "animal"
- line 58: change "doppler" to "Doppler"
- lines 87-95: what was the method of determining end-systole and end-diastole? Was it ECG guided? If yes, it should be clearly stated and the method of ECG recording should be briefly described; if not, it should be explained why the Authors did not use ECG during echocardiography and what was the method of determining measurement points; furthermore, the Authors used standard echocardiographic measurements, but it would be valuable to cite appropriate guidelines
- lines 112-116: was the data normality tested? If yes, please specify the method; if no, please conduct the analysis and compare and present data accordingly to normal and non-normal distribution of each parameter;
- lines 112-116: a comparison of parameters between sexes should be performed with Student-t test (or Mann-Whitney U analysis if the data are non-normally distributed) and not with ANOVA analysis
- line 117: Please add a short description of the studied population in the beginning of the "Results" section, i.e. age, body weight, sex distribution in the whole group and each species / hybrids
- line 118: Table 1 presents not only Doppler echocardiographic parameters but all echocardiographic parameters
- Table 1: a quotation mark in mean +/- standard deviation [minimum-maximum] is unnecessary
- Table 1: data should be presented either as mean +/- standard deviation, or as median [minimum-maximum], or as median [interquartile range], depending if they show normal or non-normal distribution
- Table 1: provide exact p-values for comparisons presenting differences (presently marked with a,b superscripts)
- Table 1: add information about a number of animals presenting mitral / tricuspid regurgitation and add the regurgitation results to the table
- lines 127-128: This sentence belongs to Materials and Methods section (together with Figure 3)
- lines 133-137: I believe these are abbreviations for the figure, but in this format it is unclear
- lines 146-147: it is unclear what kind of subtitle is presented
- Results section: in my opinion performing correlation analysis and adding information about correlations of examined parameters and animals' age and body weight would be valuable (either in the whole group or separately for each species / hybrids)
- Discussion section: please broaden the section by discussing the obtained results in the light of values obtained for other monkey species; some of the parameters should and would be body size-related, while some should not be.
- lines 156-158: rephrase the sentence to avoid repetition
- lines 166-172: please discuss the possible reasons of observed differences as compared to the literature
- lines 173-174: the sentence is unclear - please rephrase
- line 183: what do you mean by "mean" in the first parenthesis? Is it a mean value for healthy animals? it is not clear
- line 186: please change "necroscopy" to "necropsy"
- lines 179-186: it is not clear if the animal with DCM phenotype was originally included in the study? If yes, what was the duration between the study and the presence of clinical signs and did the animal show normal results during the study? Please clarify.
Round 2
Reviewer 1 Report
Comments and Suggestions for Authors
Dear authors,
Thank you for taking into accounts my suggestions. However, several details are still left:
simple summary:
Line 15: i think ‘for using’ should be ‘by using’
Line 17: cardiac reference valuer were assessed can be deleted as it is now already mentioned earlier.
Abstract:
Line 22: delete “, and as experimental models for diseases’.
Line 22/23: Add ‘Currently, echocardiographic reference values are missing. Therefore,….
Line 25: replace device by ultrasound or veterinary ultrasound (depending on what you used)
Line 27: delete ‘with an average ……of age’. Not needed to provide this here.
Line 31: diameter without a s
Line 32: ‘There’ looks misplaced.
Line 34/35: i prefer the line from the simple summary: These data serve as a refence for monitoring cardiac health in marmosets, aiding in both conservation and the management of captive primates.
Introduction.
- Non-human primates (NHP and in line 42 ‘on several species of NHP’. Change both to marmosets (callithrix species).
- Delete lines 48-54. The explanation of this species and locations has no value in this story.
Material and methods:
Line 68: 5.3± 3.2 is not correct represented as data seems not gaus distributed. Mean average ±2SD should contain 95% of the data but 5.3 minus 2x- 3.2= negative. Better option: an average of 5.3 years (range 0.4 to 15 years). And then line 64: ‘which ranged from 0.4 to 15 years’ can be deleted.
Line 78: were catched by the keepers with ‘ a net’ ‘ hand’ ‘by….’
Line 78: ‘After, there…’ should be ‘subsequently, the animals were…’
Line 79: any idea of the used flow e.g., 0.3 L/minute?
Line 81: was this machine a veterinary one?
Line 92: figure legend is incorrect with A, 1B. Should be A. what we see B. what we see. Make fort he reader clear what is the difference.
Line 97: diameter -s
Line 122: i think it should be flow -s
Results
Line 137: in line 129 you already used SD so use this abbreviation here too
Line 146-149: Is this text or figure legend? Confusing! If this is not a figure legend, please rephrase to text style.
Line 152: a verb is missing in this sentence e.g., was observed
Discussion
- In line 160-170 you describe the values of other species. Please add if those reports are in line what one would expect with different bodyweight.
- In line 158/159 you mention rhesus, spider, capuchin and cyno. In line 161-170 no cynomolgus data is mentioned. But in line 163-164 you mention marmoset and in line 166-167 callithrix. This is confusing. Is cyno missing? Is capuchin missing? Is the marmoset your own data?
- Line 172-174: with this sentence the discussion should be started. Line 175-176 should be moved tot he bodyweight section paragraph as there a bodyweight conclusion is missing.
- In line 180 you use the abbreviation HR while in line 126 you could already implement this. Please use the HR from line 126 or don’t use it as > line 180 you don’t use the abbreviation.
- Line 180-184: far too long sentence which has too many verbs. Please rephrase. And found could be described and the ‘earlier studied’ should be followed by reference.
- Line 194: you write 6.1±3.1 while in the introduction you wrote 5.3±3.2 Is one wrong? I advice to change, as the data is not gaus distributed, to : average age of 6.1 years (range…..)
- Line 200: when you write down ‘veterinary care’, the readers want to know what kind! Heart medication?
- Line 207: safer? I am not sure if that is correct as most inhalation anesthetics cause dose-dependent depression of the cardiovascular system so the safeness depends on the skills of the anesthetist. Therefore, I would suggest to delete line 206-209.
- Line 204: ten instead of 10 and C.jacchus should be intalics
- Line 210: Maybe change to: after completing the study, a 15-year-old female pennicilate was presented to the veterinarian with abdominal distension and anorexia.
Reviewer 3 Report
Comments and Suggestions for Authors
I wanted to thank Authors for all the replies and addressing my concerns.
I have no further comments.
Congratulations on your work!
Author Response
We appreciate your considerations.
Round 3
Reviewer 1 Report
Comments and Suggestions for Authors
Dear,
thank you for implementing my suggestions. I would only suggest to delete:
The genus Callithrix includes six species described in Brazil: Callithrix aurita (É. Geof- 47
froy in Humboldt, 1812), Callithrix flaviceps (Thomas, 1903), Callithrix geoffroyi (É. Geoffroy 48
in Humboldt, 1812), Callithrix jacchus (Linnaeus, 1758), Callithrix kuhlii (Coimbra-Filho, 49
1985), Callithrix penicillata (É. Geoffroy, 1812) [5,6]. Marmosets can be found throughout 50
Brazil [7].
I think this has no additional value.
Further okay, well done.
Author Response
Comments 1: Ok (deleted).
We appreciate your considerations.